# Bullying in the Arab World: Definition, Perception, and Implications for Public Health and Interventions

**DOI:** 10.3390/ijerph21030364

**Published:** 2024-03-19

**Authors:** Muthanna Samara, Nura Alkathiri, Mahitab Sherif, Aiman El-Asam, Sara Hammuda, Peter K. Smith, Hisham Morsi

**Affiliations:** 1Department of Psychology, Kingston University London, Penrhyn Road, Kingston upon Thames, London KT1 2EE, UK; 2Department of Psychology, Goldsmiths College, University of London, London SE14 6NW, UK; 3National Centre for Cancer Care and Research (NCCCR), Hamad Medical Corporation (HMC), Doha 1705, Qatar

**Keywords:** bullying, Arabic, Arab, terminology, definition, perception, attitude, culture, aggression, violence, Middle East

## Abstract

The present research aimed to examine bullying among diverse Arab nationalities residing in Qatar across two separate studies. Study 1 examined how Arabic-speaking adolescents and adults describe and perceive bullying, participants (*N* = 36) from different Arab nationalities (i.e., Egyptians, Qataris, Syrians, and other Arabs) were presented with three tasks in a focus group where they were asked questions about how they describe and perceive three scenarios without reference to the term “bullying”. Findings indicated that (1) the majority of participants referred to the intention to cause harm and the imbalance of power in their descriptions, and (2) differences in describing the behaviours in the scenarios were notable when comparing Egyptians with the three other nationalities. Overall, participants frequently chose different Arabic terms (e.g., Ta’adi (تعدي)) in their descriptions of the scenarios. Interestingly, the term Tanammor (تنمُّر), which has been used in previous studies as the Arabic term for bullying, was chosen the least by the current sample. Study 2 examined how Arab-speaking students (*N* = 117) describe bullying behaviour in seven scenarios using Arabic and English terms. The procedure was administered in English in the international schools, and Arabic in the independent schools. English-speaking students often used the term “bullying”, whereas Arabic-speaking students often used behavioural descriptions (e.g., the term “solok sayea” (سلوك سيء) which translates to “bad behaviour”). These findings are discussed in relation to the definition and perspective of bullying among Arabic speakers. There is a need for further investigations to introduce a novel term for bullying within the Arabic language while considering cultural values, norms, and beliefs. This has the potential to promote heightened awareness and comprehension, enabling the formulation of customised intervention approaches, policies, and educational initiatives intended to prevent and alleviate bullying behaviours.

## 1. Introduction

Bullying among school children is a problem of growing international concern affecting social development, health, academic achievement, and well-being [1,2,3,4]. Studies carried out among school-aged children across different countries suggest that bullying is associated with social, emotional, and health problems for both the victim and the perpetrator. Some studies have also found that the adverse effects of bullying persist into adulthood [5,6,7]. Consequently, there has been a substantial expansion of scholarly work on this behavioural phenomenon over the years. The primary objective has been to gain a comprehensive understanding of its nature and devise interventions to mitigate its adverse effects.

Although different terms have been used to define bullying across different languages, the current consensus in the literature is that bullying is a repeated aggressive behaviour perpetrated by one or more people to intentionally inflict physical or psychological harm and involves an imbalance of power, leaving the victim in an almost defenceless position [8]. This disparity of power may be direct-physical (e.g., assault), direct-verbal (e.g., threat), and/or relational (e.g., social exclusion). While these traditional forms of bullying have long since been considered a problem among school children, the exponential growth of online social networking has led to a new form of bullying that exists online. Cyberbullying is defined as a repeated and intentional aggressive act perpetrated by one or more people using electronic forms of contact against a victim [9]. There is now widespread use of Olweus’s bullying criteria in questionnaires and related survey instruments in order to understand bullying characteristics and measure normative prevalence. A modified version of Olweus’s questionnaire was also used across five countries, including England, Japan, Norway, the Netherlands, and the United States [10,11,12].

While there are a few cross-national studies, the existing literature shows considerable variability in the prevalence of bullying across countries [1,13]. An international survey of adolescent health-related behaviour showed that the percentage of students who reported victimisation on at least one occasion ranged from 15% in some countries to 70% in others [13,14,15]. Similarly, the prevalence of frequent bullying reported internationally was significantly higher in some countries (e.g., Malta) compared with others (e.g., Ireland) [13,16,17]. A study also found that adolescents in Baltic countries reported a higher rate of bullying compared with those in northern European countries [1].

Bullying is thus common and exists in all cultures and countries [18]. However, there is a lot of variation in bullying prevalence and nature [19]. Some of these discrepancies are related to different cultural contexts and how bullying is perceived in these cultures. Specific methodological research approaches in Western countries have dominated this area. For example, the Olweus questionnaire has been translated into different languages around the world to assess and compare prevalence rates. However, there is ultimately a wide range of methodological flaws with this approach, which indicates that researchers should treat findings with more caution. These include cultural variations between different nationalities and/or ethnicities [20]. There are also variations in research with no specific definition and/or terminology [21] and differences in the application of research methodologies [22], methods of data collection (e.g., self-report, interviews), response measures (i.e., scales and response time), types of statistical data analyses [20,23], and cultural attitudes. Thus, it is important to investigate bullying while using the cultural context as well as the correct norms and values. These are vital issues that need to be considered in any national or international research on bullying. The meaning of the word “bullying” as defined by Olweus [24], varies across research and applied settings among children. This could have different implications for how to deal with and intervene with bullying.

The main implication of cross-national comparisons is the comparability of terminology. Arora [25] identified a list of terms that are cognate with the word “bullying” in both the English language and several other languages. This suggested a need for a universal measure that consists of the same core concepts across different countries to facilitate valid comparisons [2]. Smith et al. [26] found that terms from different languages that correspond to the English word “bullying” evoked different meanings about the type of bullying that was reported by 8- to 14-year-old students. This led to the suggestion that both language and culture are strong determinants of children’s understanding of bullying. Subsequent studies on this topic led Smith and colleagues to conclude that the assessment of bullying in cross-national research should be consistent across languages and culturally inclusive [18].

While the current literature shows the existence and impact of bullying across the world, research shows clear discrepancies in how the term “bullying” is interpreted across different languages. Prevalence rates are significantly high in developing countries, despite the limited bullying research being undertaken in these regions [27]. Bullying research is also very scarce in Arabic-speaking countries. One study that examined bullying in the Arab world reported an average prevalence rate of 34.2% from 19 countries situated across four world regions (Africa, America, Asia, and the Middle East). This included individual prevalence rates from five Arab-speaking countries: 44.2% for Jordan, 31.9% for Morocco, 39.1% for Oman, and 20.9% for the United Arab Emirates [28].

However, the existing literature from the Middle East has some limitations. First, the issue of comparability of terminology is evident in some of these studies. In a study exploring bullying behaviour in children and adolescents, two different Arabic terms (“tanammor, تنمر ” and “esteqwa, استقواء”) were used intermittently to refer to bullying [29]. The first term (“tanammor, تنمر”) refers to the word “tiger” as a symbol of physical violence, while the last term (“esteqwa, استقواء”) refers to the perception of strength and control over others. Both terms may also carry positive connotations and therefore may not always define bullying as a negative act. Second, while a range of questionnaires and survey instruments are available to measure bullying in Western cultures, very little attention has been paid to measuring bullying in the Arab world. As the development of measures to assess bullying is limited in this region, translated questionnaires are often used to assess bullying in Arabic-speaking students [30,31], despite the evidence of cultural and language-based differences. While it is unrealistic to match terms across languages, it is necessary to know how comparable terms are and, if they are different, where the discrepancies exist (e.g., physical/psychological, direct/indirect, individual/group) [2]. An accurate representation of bullying is determined by the effectiveness of the method of assessment (i.e., questionnaire).

In an effort to control for cultural and language-based differences in the assessment of bullying, Taki et al. [32] developed the Pacific-Rim Bullying Measure, where children from five countries (Australia, Canada, Japan, South Korea, and the United States) were asked to describe the characteristics of bullying behaviours that they had experienced without using the term “bullying”. Using this measure across the five countries, Taki and colleagues defined bullying using the three criteria from Olweus’ widely used definition: intention, repetition, and imbalance of power. To date, this measure has not been used in Arabic-speaking countries. In addition, the Arabic language comes in different forms and dialects. The formal modern standard Arabic language, is often spoken in professional settings (e.g., books, literature, and news stations) across all countries in the Middle East and North Africa. However, variations exist between the different Arabic countries in the specific dialect used, and these are more common compared to the formal modern standard Arabic language. Due to colonial history dialects vary and are sometimes even mixed with foreign languages such as French in Lebanon, Morocco, Tunisia, and Algeria, and English in Egypt and the Gulf countries [33].

This research paper addresses the dearth of published work on bullying in the Middle East by conducting two comprehensive studies aimed at examining the understanding and description of bullying-related behaviours among Arabic-speaking children and adults. Additionally, this paper is the first to consider variations in the terminologies used for bullying within the Arabic language. Building on previous research that suggests limited comprehension of the term “bullying” and its cultural nuances, we anticipate identifying diverse terminologies associated with bullying, with a focus on specific behaviours rather than an overarching term. Furthermore, we expect variations in terminologies across different Arabic-speaking countries due to contextual influences. Two distinct studies were conducted: (1) An investigation into how Arabic-speaking participants describe and perceive bullying behaviours; and (2) An exploration of how Middle Eastern participants describe bullying behaviours using both Arabic and English terms. These investigations were performed while considering three key variables: gender, nationality (Egyptian, Qatari, Syrian, Palestinian, and other Arab), and language (Arabic and English). Therefore, the research questions of this study are:Is there a lack of standardised terminology for bullying across Arabic-speaking Middle Eastern populations, and to what extent does this variation in terminology result from limited understanding and cultural differences? How do participants use various words and phrases to describe bullying behaviours instead of a singular, universally accepted term?To what degree do descriptions of bullying behaviours in Arabic-speaking Middle Eastern populations tend to be linked to specific acts and actions rather than encompassing the broader concept of bullying itself? How does the understanding of bullying relate to concrete experiences and actions rather than a comprehensive definition?In what ways do variations in the terminology used to describe bullying behaviours exist across different Arabic countries and nationalities within the Middle Eastern region? How do these variations reflect distinct cultural norms, perceptions, and experiences related to bullying? To what extent does nationality shape perceptions and descriptions of bullying behaviours, influenced by unique socio-cultural factors within each country?When comparing Arabic and English terms for describing bullying behaviours, how do participants exhibit more nuanced descriptions when using their native Arabic language compared to English? To what extent are these differences in expression attributed to variations in linguistic expression and cultural nuances between the two languages?How do perceptions and descriptions of bullying behaviours differ across genders in Arabic-speaking Middle Eastern populations? To what degree are these differences influenced by societal expectations, gender norms, and experiences related to bullying?

The proposed research aims to contribute to the existing literature on bullying, advance the understanding of culturally specific terminologies, and shed light on the complex dynamics of bullying within the Middle Eastern context. These aims will be explored across three variables: gender, nationality (Egyptian, Qatari, Syrian, Palestinian, and other Arab nationalities), and language (Arabic and English). The findings from these studies will have implications for developing tailored interventions and support systems to address bullying effectively in the region.

## 2. Materials and Methods

### 2.1. Study 1

Participants from diverse Arab nationalities residing in Qatar took part in a focus group where they were asked questions about the behaviours depicted in three different bullying scenarios. This study aimed to examine how participants described and perceived the scenarios without using the actual term “bullying”. Systematic differences between participants’ gender and nationality were also explored.

#### 2.1.1. Participants

Informed consent was sought from participants and the school management where the study took place. In the case of children under the age of 16, consent was also sought from their parents or legal guardian. A total of 36 participants between 9 and 50 years old (*n*: 20 participants are 18 years old or below with their families; *n*: 16) were recruited from different schools in Qatar that consisted of individuals from a variety of different Arab nationalities (see Table 1 for demographic data). The sample included participants who agreed to take part in the study, were recruited based on convenient sampling, and were of various ages.

#### 2.1.2. Design

This is a qualitative study that relies on a focus group as a method of data collection. This method is well-established for its rich and detailed information about participants’ perceptions, beliefs, and attitudes towards a particular topic. On this basis, this method was used to gain an understanding of the terms that participants use to describe behaviours that represent the characteristics of bullying. This qualitative approach was employed to control for cultural and language-based differences that may occur during the study.

#### 2.1.3. Materials

This study consisted of three scenarios that represented the characteristics of bullying behaviour. The scenarios were presented in written form on a computer screen. The scenarios included:Every day at school, Reem steals Heba’s chocolate bar taken off her from her lunchbox. She also sometimes takes her money.He doesn’t take things from any other people in the class; and Mohamed and Haitham regularly and anonymously spread false stories about one of their colleagues in class through Facebook. They do this with the intention of harming him.Mohammed punched Faisal in the face during the lunch break at school, leaving him with a bleeding mouth.

Once participants read each scenario, they were presented with three separate tasks that aimed to explore how they defined bullying:Participants were asked to provide a term that best described the action being represented in each scenario.Once they had completed the first task, participants were asked four further questions that explored their perceptions about the behaviours in the three scenarios: (1) What do you generally think about the previous scenarios? (2) Are these scenarios or similar ones acceptable/unacceptable, and why? (3) Have you witnessed or experienced similar situations before? (Would you like to share any of it?) (4) How did the experience make you feel? How was it for you to go through such a situation?Participants were then presented with a 7-item word list that consisted of terms that described the characteristics of bullying (see Table 2). Participants were asked to choose a term from the list that best described the three scenarios. These terms were chosen from the answers that the participants produced from the first tasks while adding other common words that are used in Arabic literature about bullying (e.g., tanammor/تنمر—bullying). Participants were also given the option to use their own terms.

#### 2.1.4. Procedure

Divided into small groups of 10, participants took part in focus group interviews in a quiet room where they were seated comfortably. Participants were informed that the interview would be audio recorded and were briefed about the study without referring to the term “bullying”. Participants were then shown the three scenarios on a screen and were asked to verbally state a term that best described the behaviour as each scenario was presented.

Once participants provided their verbal descriptions of all three scenarios, they were presented with four questions that examined their perception. Participants were given a few minutes to provide their answers as each question was presented.

Finally, participants were asked to select a word that described each scenario from a list of pre-determined terms presented on an A4 sheet of paper.

At the end of the focus group interview, participants were debriefed about the purpose of the study and provided with the opportunity to ask questions and/or raise any related issues.

### 2.2. Study 2

Participants from international and independent schools in Qatar participated in different focus groups where they were asked to give a term that explains the behaviours in 7 scenarios. The focus groups were administered to students in English in the international school and in Arabic in the independent schools to examine language-based differences. Differences between the four grouped nationalities were explored: Egyptians, Qataris, Syrians, and other Arab nationalities, which represented the minority students in Qatari schools. Syrian students were only found in independent schools.

#### 2.2.1. Participants

A total of 117 participants (70 males, 59.8%, and 47 females, 40.2%) between 11 and 15 years old took part in the study. Participants were recruited from international schools (*n*: 61, 52.1%) and independent schools (*n*: 56, 47.8%) and represented diverse Arab nationalities (see Table 3 for demographic data).

#### 2.2.2. Materials

This study consisted of 7 scenarios, 6 of which represented the criteria of bullying behaviour:A group of girls are making fun of one of the girls at school; they often call her bad names.Three friends in the same class repeatedly send Faisel text messages calling him names (impolite). Faisal is upset by this but does not say anything to anyone.A boy/girl keeps receiving a message on his/her computer from an unknown person. The messages are often strange and say things such as ‘I hate you’.Over the last three weeks, a student has been repeatedly hit by other students.Hasan once joked with Ahmad, but Ahmad got angry and punched him in the face.A group of friends are telling untrue stories about one of their colleagues; they have done that many times.Mohamed and his friends never let Naif play with them.

Scenario one reflects direct verbal bullying, scenarios two and three reflect cyberbullying, scenario four reflects direct-physical bullying (traditional bullying), while scenarios six (rumours) and seven (exclusion) reflect relational bullying. One scenario (scenario 5) did not depict bullying behaviour and was used as a control. Following the presentation of each scenario, participants were asked to produce a term that explained the behaviour presented.

#### 2.2.3. Procedure

Participants were briefed about the study without referring to the term “bullying” prior to the start of the focus group interview. Participants were given a few minutes to read each scenario while seated in front of a computer screen. They were then asked to verbally state the term that describes the behaviour presented in each scenario.

### 2.3. Validity and Reliability

The materials and scenarios were constructed in English by three expert researchers in the field of bullying (authors: Muthanna Samara (MS), AEA and PKS). Two of them are fluent in the Arabic language as their mother tongue (MS and AEA). The scenarios were translated to Arabic and back translated to English, and they were compared to ensure compatibility and validity. Based on the criteria of bullying (repetition, intentionality, and imbalance of power), ecological validity was also checked to ensure that the scenarios are suitable to be used culturally inside Qatar. This was conducted in consultation with a group of 15 psychologists who work in schools in Qatar.

## 3. Results

### 3.1. Results for Study 1

Content analysis was carried out to examine the results from the three separate tasks. These results are presented below:

#### 3.1.1. Task 1

Table 4 shows the frequency of the terms that the participants used to describe the behaviours being depicted in the three scenarios. The overall findings revealed that for scenario 1 the terms “jealousy” and “theft” were used more frequently (*n*: 6, 16.6%). For scenario 2, the term “jealousy” was the most common answer (*n*: 8, 19.4%), while the term “aggression” was more often used to describe the actions in scenario 3 (*n*: 6, 16.6%).

#### 3.1.2. Task 2

In the second task, participants were asked four further questions that explored their perspective on the behaviours in the three scenarios. The interrater reliability between the two researchers who did the content analysis separately and independently was 0.81.

##### Qualitative Analysis for Responses to the First Question

In analysing responses to the first question, none of the participants referred to the behaviours in the three scenarios by the three criteria of bullying as (1) Repeated over time, (2) With the intention to harm, and (3) With an imbalance of power.

Repetition: Participants did not refer to the repetition of the behaviour in their response to the first question.

Intention to cause harm: Seven (19.4%) participants referred to the intention to cause harm in their explanation of the behaviours in the three scenarios. More specifically, one participant stated that such behaviours are “*attitudes or acts from one person to another with the intention to bother or harm them*”. Participants used a range of terms to indicate their intention to cause harm, including “*assault*” and “*physical violence*”.

The majority of the participants who referred to the second criterion in their answers were females (*n*: 7; 85.7%). Slight nationality-based differences were also found in participants’ responses to this question: 14.2% were Egyptians, 42.9% were Qataris, 14.2% were Syrians, and another 14.2% were from other Arabic nationalities.

Imbalance of power: Six participants (16.6%) from the total sample referred to this characteristic in their response to the first question. The imbalance of power was also mentioned through narratives that refer to the victim as being “*weak*”, as one participant stated, “*feeling strong and using the strength against the weaker person in the equation*”.

The majority of the participants who referred to the imbalance of power in their response to this question were again female (*n*: 4, 66.7%). In terms of nationality, 33.3% of the participants who referred to this criterion were Syrians (*n*: 2), 50.0% were Qataris (*n*: 3), and 16.7% were from other nationalities (*n*: 1). Egyptian participants in the present sample did not mention this criterion in their answers.

##### Qualitative Analysis for Responses to the Second Question

Participants were asked to state whether they found the behaviour exhibited in the scenarios acceptable. Most of the participants (*n*: 26) described the behaviours from the three scenarios as “*unacceptable*” (72.2%). Five participants (13.9%) went on to elaborate on the limits of what makes these behaviours acceptable and explained that these scenarios might be “*acceptable*”, “*normal*”, *and* “*fun*” if they occur between friends. For example, one participant stated that “*between friends, it is acceptable if they are ok with it and considered it as having fun*”. One participant also reported that this behaviour is unacceptable once it “*intrudes over the other’s rights*”.

Interestingly, two out of the three criteria from Olweus’s definition of bullying (repetition and intention to harm) were used by the present sample to determine whether the behaviour in these scenarios was acceptable. Overall, 13 participants (36.1%) stated that behaviours that are repeated and/or intend to cause verbal and/or physical harm are unacceptable. One participant stated, “*If these behaviours are repeated over and over or with the intention to harm others, then it’s unacceptable, but if it happens just once in a while, then they might be just playing because this is how children play*”.

Two participants highlighted the differences in how Qatar and Western societies deal with unacceptable behaviours among school-aged children. Both participants explained that such behaviours are not dealt with accordingly and often go “*unnoticed*”. This is described in more detail in one participant’s answer: “*In the US or the UK they solve these situations and they know how to deal with similar behaviours but here in Qatar they do not know what to do and I don’t think that teachers or parents are aware of what’s happening because the students are scared to tell their parents*”.

Five participants (13.9%) mentioned in their answers the right to self-defence against behaviours such as those depicted in the three scenarios, emphasising that the victims should defend themselves. This argument is clearly demonstrated in one participant’s answer: “*A person should defend himself, if a kid was hit or pushed by another kid then he has the right to push him back, as it’s not acceptable not to claim your right*”. Another participant stated, “*What is unacceptable is not to defend yourself, when you allow others to harm you*”.

As a considerable majority of participants identified the behaviour as unacceptable, there were no gender or nationality differences found in relation to this question.

##### Qualitative Analysis for Responses to the Third Question

Participants were asked to provide examples of incidents that they may have experienced or witnessed that are similar to the behaviours described in the three scenarios.

Some participants reported accounts of incidents that were either personal or experienced by a close relative or friend. Participants’ reports were grouped into six categories that represent the most common behaviours experienced: theft, verbal bullying, physical bullying, relational bullying, rumours, and gossiping.

*Theft*: Eleven participants (30.6%) from the total sample reported incidents of theft occurring among students at school. Participants claimed to have lost a range of personal items, including money, food, and school stationery. One participant claimed that “*At school, I used to come back from break I used to find all my pencils are gone this happened every day and I consider this as theft/stealing*”. Another participant reported, “*I had my money stolen from me regularly at school*”.

Findings also showed that 63.6% of those who reported experiencing or witnessing theft were female (*n*: 7). In terms of nationality, 36.4% were Egyptians (*n*: 4), 36.4% were Qataris (*n*: 4), 9.1% were Syrians (*n*: 1), and 18.2% were from other nationalities (*n*: 2).

*Verbal bullying:* Nine participants (25.0%) from the total sample described incidents that depicted the characteristics of verbal bullying using terms such as “*insulted me*”, “*make fun*”, “*saying very bad stuff*” and “*throwing words at me*”.

Overall, 88.9% of the participants who reported either experiencing or witnessing verbal bullying were female (*n*: 8). Across nationalities, 66.7% were Qataris (*n*: 6), 12.5% were Egyptians (*n*: 1), and 22.2% were from other nationalities (*n*: 2).

*Physical bullying*: Thirteen participants (36.1%) described incidents that depicted the characteristics of physical bullying, where there was a direct attack that purposefully caused bodily harm. These participants described their account using terms such as “*hit*”, “*fight*”, “*pulled my chair and let me fall*” *and* “*push him*”. For example, one participant reported that “*I started to hit them then they hit me back so I started to hit them harder until they stopped*”. Another participant admitted to resorting to physical violence to defend herself, claiming that “*she used to annoy and bother me all the time, and after I hit her, she apologized to me*”.

Furthermore, three participants (8.3%) reported hitting/fighting to stop the perpetrator from physically or verbally attacking them. One participant admitted to advising his daughter to use brute force to stop the perpetrator. He stated, “*when my daughter was in one of the independent schools, there was a girl who used to bother her all the time, so I asked my daughter to hit that girl because she is the bully of the class*. *My wife was against that because she came from an American background and I told her that this is how we solve things here in Qatar and we’re no longer in the US*”.

Sibling rivalry was also reported in some participants’ accounts. Four participants (11.1%) claimed that they had witnessed fighting among children among their close relatives. For example, it was stated that “*from my own experience I have two nephews the youngest is very loved and old boy is always jealous of his young brother so he tries to push him, hit him, trick him and I think this is out of jealousy*”.

Another participant explained that such behaviour was common among her own children, as she stated that “*the old brother always preventing his young brother to watch his favourite programs on the TV*”. “*Also the old brother takes his young brother’s toys and play with them alone without sharing and he gives it back when he’s done*”.

Further analysis showed that half of the total male sample (*n*: 10, 50.0%) reported either experiencing or witnessing physical bullying, compared with only seven female participants (*n*: 25, 23.1%). Among the male participants who reported physical bullying, 10.0% were Qataris (*n*: 1), 20.0% were Egyptians (*n*: 2) and 20.0% were from other Arab nationalities (*n*: 2). For the female participants, 57.1% were Qataris (*n*: 4), 14.2% were Syrians (*n*: 1), and 14.2% were from other Arab nationalities (*n*: 1).

*Relational bullying:* Only 2 participants (5.6%) from the total sample reported experiencing social exclusion from their peers at school. This is evident from one who stated, “*Last year I had a very close friend and by the end of the year she started to talk to another girl and started to exclude me, I don’t think she means it because sometimes they have some activities to do alone without me, so I started to sit with other friends*”. Social exclusion was only reported by two female participants.

Another participant referred to rumours and gossip at school, as she claimed that spreading false stories was very common.

*Online bullying*: Direct online bullying was not reported in the present sample. However, one participant reported witnessing a member of her peer group delete another girl from her online social networking circles following an earlier argument that erupted between them. The participant stated that “*a girl once insulted another girl, so she reacted by not following her on Instagram and made the other girls do the same until I saw the first girl breaking her iPhone because she was very angry*”.

##### Qualitative Analysis for Responses to the Fourth Question

Participants were asked about how the incident they had experienced or witnessed made them feel.

In their accounts of the incident, 4 participants (11.1%) expressed that the incident made them feel “*stress*”, “*sadness*”, “*hatred*”, “*embarrassment*”, *and* “*feeling bad*”.

Upon reflection of the incident, one participant stated, “*I used to feel extremely bad, and when I started to react I wasn’t loved even by the teacher and it was hard not to have the teachers on my side*”.

The emotional consequences of bullying were further explored by a participant who stated that “*I had a peer who used to hit me every day until one time I snapped and the guy stopped hitting me anymore, then I realized how this behaviour puts you down*”.

One participant postulated that the perpetrators of bullying may suffer from problems as he stated, “*I believe that the bully has problems that are reflected in bullying others, or he has been through a situation that made him a bully, and the environment that he came from is unstable that made a bully out of him*”.

#### 3.1.3. Task 3

For the final task, participants were asked to choose a term from a seven-item word list that best described the actions in the three scenarios:Every day at school, Reem steals Heba’s chocolate bar taken off her from her lunchbox. She also sometimes takes her money.He doesn’t take things from any other people in the class; and Mohamed and Haitham regularly and anonymously spread false stories about one of their colleagues in class through Facebook. They do this with the intention of harming him.Mohammed punched Faisal in the face during the lunch break at school, leaving him with a bleeding mouth.

Table 5 shows the frequency of the participants’ answers. The majority of participants (31.6%, *n*: 12) chose the term “dominance” (“Tassalot’, تسلط”) to describe the behaviour in scenario 1. In contrast, the term “harm” (“Az’a, أذى”) was more frequently chosen (40%, *n*: 16) for scenario 2. For the final scenario, most participants (20.5%, *n*: 8) chose the terms “harm” (“Az’a, أذى”) and the Egyptian word for bullying, (“Blatag’ah, بلطجة”), which is related to gangsterism. Overall, the terms trespass (“Ta’adi, تعدي”) (71.7%; *n*: 28) and harm (“Az’a, أذى”) (68.4%; *n*: 28) were more frequently chosen across all of the three scenarios. Interestingly, the term “Tanammor”, which has been used in previous studies as the Arabic definition of bullying, was chosen the least by the current sample (see Table 5).

Table 6 shows the frequency of the terms chosen from the 7-item word list for each of the three scenarios across the four groups of nationalities. Findings show that the majority of Egyptians (33.3%) and Syrians (50%) chose the term “dominance” (“Tassalot’, تسلط”) to describe scenario 1, whereas Qataris (33.3%) and students from other nationalities (36.4%) repeatedly chose the term “trespass” (“Ta’adi, تعدي”). For scenario 2, the term “harm” (“Az’a, أذى”) was repeatedly chosen across all four nationalities: Egyptians (22.2%), Qataris (41.7%), Syrians (75%), and other nationalities (54.5%). For scenario 3, the term “gangsterism” (“Baltag’ah, بلطجة”) was more frequently chosen by Egyptians (55.6%) compared with the other three groups: Qataris (8.3%), Syrians (25.0%), and others (9.1%). In contrast, the term “harm” (“Az’a, أذى”) was only used by Qataris (33.3%) and was the most frequently used by participants from this nationality in describing scenario 3.

### 3.2. Results for Study 2

The focus group results were analysed and are presented separately for the international and independent schools:

#### 3.2.1. Results from Qatar International Schools

Table 7 shows the frequency of the terms used to describe each scenario among participants in international schools.

Scenario 1: The term “bullying” was the most frequent term used to describe the behaviour depicted in this scenario (47.5%). This term was often used by students from other nationalities (62.1%, *n*: 18), followed by Egyptians (24.1%, *n*: 7) and Qataris (13.8%, *n*: 4). This term was more popular among girls (58.6%, *n*: 17) than boys (41.4%, *n*: 12).

Scenario 2: “Cyberbullying” was the most popular term used to describe this scenario (29.5%). This term was more frequently used by students from other nationalities (66.7%, *n*: 12) and Qataris (27.8%, *n*: 5) compared with Egyptians (5.6%, *n*: 1). Over half of the participants who used this term were girls (55.6%, *n*: 10). A total of 8 boys used this term to describe the scenario (44.4%),

Scenario 3: The term “cyberbullying” was also the most frequently used in this scenario (27.9%). The participants who used this term were often students from other nationalities (58.8%, *n*: 10), followed by Qataris (29.4%, *n*: 5) and Egyptians (11.8%, *n*: 2). This term was slightly more common among girls (52.9%, *n*: 9) than boys (47.1%, *n*: 8).

Scenario 4: The majority of the participants described this scenario using the term “bullying” (27.9%). This term was often used by students from other nationalities (41.2%, *n*: 7) and Qataris (35.3%, *n*: 6) compared with Egyptians (6.5%, *n* 4). A higher proportion of girls used this term (70.6%, *n*: 12) compared with boys (29.4%, *n*: 5).

Scenario 5 (Control): The participants often referred to the behaviour in scenario 5 as an inappropriate joke (26.2%). Over half of the participants who used this term were from other nationalities (62.5%, *n*: 10). Qataris and Egyptians (16.7%, *n*: 3) also equally referred to this term in their descriptions of this scenario. This term was more frequently used by boys (56.3%, *n*: 9) than girls (43.8%, *n*: 7).

Scenario 6: A high proportion of participants described the behaviour in this scenario using the term “gossip/rumours” (37.7%). Students from other nationalities often used this term (47.8%, *n*: 11), followed by Qataris (34.8%, *n*: 8) and Egyptians (17.4%, *n*: 4). A higher proportion of girls (56.5%, *n*: 13) used this term to describe the scenario compared with boys (43.5%, *n*: 10).

Scenario 7: “Left out” was the most common description of the final scenario (9.8%). Half of the participants who used this description were from other nationalities (50.0%, *n*: 3), while the other half of the participants were 2 Qataris (33.3%) and one Egyptian (16.7%). This term was equally common among both girls and boys (50.0%, *n*: 3).

#### 3.2.2. Results from Qatar Independent Schools

Table 8 shows the frequency of the Arabic terms used to describe each scenario among participants from independent schools.

Scenario 1: The term “mockery” (“Istihzaa, إستهزاء”) was the most frequently used to describe this behaviour among the students (51.8%). Over half of the participants who used this term were Qataris (55.2%, *n*: 16). The remaining number of participants who used this term were from other nationalities (27.6%, *n*: 8), Syrians (10.3%, *n*: 3), and Egyptians (6.9%, *n*: 2). A significant majority of participants who used this term were boys (72.4%, *n*: 21) compared with girls (27.6%, *n*: 8).

Scenario 2: The majority of the participants (16.1%) described this action using the term “bad behaviour” (“Solok sayea’, سيء سلوك”). This term was very popular among Qataris (77.8%, *n*: 7). A small proportion of students from other nationalities (22.2%, *n*: 2) also used this term. A high proportion of participants who used this term were boys (66.7%, *n*: 6) compared with girls (33.3%, *n*: 3).

Scenario 3: The term “bad behaviour” (“Solok sayea’, سيء سلوك”) was also common in this scenario (16.1%). This term was often used by students from other nationalities (55.6%, *n*: 5), followed by Qataris (22.2%, *n*: 2) and Syrians (22.2%, *n*: 2). This term was only used by boys to describe the scenario (100.0%, *n*: 9).

Scenario 4: Perception of strength and control upon others (“Istiqwaa’, إستقواء”) was the most common term (12.5%) used to describe this scenario. This term was equally popular among students from Qatar and other nationalities (42.9%, *n*: 3). One Syrian student also referred to this term (14.2%). Only boys used this term to describe the scenario (100.0%, *n*: 7).

Scenario 5 (control): The majority of the participants (30.4%) referred to the behaviour as a “heavy joke” (“Mazih Thaqeel, ثقيل مزح”). This term was often used by Qataris (42.1%, *n*: 8) compared with students from other nationalities (31.6%, *n*: 6), Syrians (10.5%, *n*: 2), and Egyptians (5.2%, *n*: 1). This term was more common among boys (89.5%, *n*: 17) than girls (10.5%, *n*: 2).

Scenario 6: The term “liar” (“Kaz’ib, كذب”) was the most frequently used to describe this scenario (14.3%). This term was common among participants from Qatar (62.5%, *n*: 5) and other nationalities (37.5%, *n*: 3). A higher proportion of boys (62.5%, *n*: 5) used this term compared with girls (37.5%, *n*: 3).

Scenario 7: The majority of the participants (32.1%) described the behaviour in this scenario with the term “hate” (“Karahiyah, كراهية”). This term was more common among Qataris (55.6%, *n*: 10) compared with students from other nationalities (38.9%, *n*: 7) and Syrians (5.6%, *n*: 1).

## 4. Discussion

The study examined how bullying is defined and perceived by different Arab nationalities, utilising two studies. The first study investigated how Arabic-speaking participants described and perceived bullying. The majority referred to the intention to cause harm and the imbalance of power in their descriptions, and there were differences in describing the behaviours between different participants from different Arab countries. The participants frequently used the Arabic term “trespass” (Ta’adi; تعدي) in their descriptions. Interestingly, the term Tanammor (تنمر), which is related to behaving like a tiger (connecting the behaviour to physical aggression) and has been used in previous studies as the Arabic term for bullying, was chosen the least by the current sample, especially amongst children and adolescents. The second study shows that English-speaking students often used the term “bullying”, whereas Arabic-speaking students often used the term “bad behaviour” (“Solok sayea’—سلوك سيء”). The findings are unique and important, as this is the first study to investigate the definition, description, and perception of bullying scenarios and related words in the Arabic Middle Eastern context. Studies in the Arab world are few and mostly investigate the prevalence of bullying [33]. In addition, studies in the Arab world vary in relation to the use of one common word in Arabic for bullying; some use the word “Tanammor” (behaving like a tiger), while others use violence, connecting all behaviours to physical acts or aggression [33].

There is a lack of studies on the relationship between cultural factors and practices, and bullying in the Arab world. The Arabic language is one of six official languages in the United Nations [34,35]. It is the official language of 27 countries and is spoken by more than 422 million people in the Arab world and more people outside of the Arab world since it is the language of the Quran, the Islamic holy book [33,35]. Boudad et al. [35] stated that “Arabic is ranked the fourth most used language and the fastest growing with a growth rate of 6091.9% in the number of Internet users”. Hence, this study is vital as a starting point, and more studies are needed to investigate bullying and its definition in the Arab world.

Different languages use different terms to define bullying across countries and even within each country; however, the current consensus in the literature is that bullying is a repeated behaviour to intentionally inflict physical or psychological harm and involves an imbalance of power [8].

As was found in this study, different Arabic terms have been used to refer to bullying. These terms do not fully refer to bullying in the same way that it is perceived for English-speaking individuals but to actual behaviours. Therefore, the question is whether the same perception is perceived by the Arabic-speaking individual to various descriptions as it is for the English word “bullying”, especially in relation to the three criteria of intention to harm, imbalance of power, and repetition. These are more notable when words such as “violence”, “aggression” and “teasing” are used as these can mostly refer to single negative behaviours and not necessarily to bullying by its comprehensive perception. In addition, translated questionnaires are often used to assess bullying among Arabic-speaking students [30,31], despite the evidence of cultural and language-based differences that affect children’s understanding of bullying [18]. A study in Lebanon [36] revealed that students referred to physical abuse as bullying. In addition, bullying was regarded as a way of having fun, of improving self-image among peers, and as a way of protecting others.

The Arabic language includes formal classical language, which is often spoken in professional settings (e.g., Quran, books, literature, news stations) across Arab countries, while variations exist in the specific dialect used in each country, and these are more common compared to the formal language. These dialects vary and are sometimes even mixed with foreign languages [33]. These vary between countries and even within the same country and depend on whether the individual lives in a rural or urban area.

By examining the impact of language on perceptions, attitudes, and behaviours within a society, we can comprehend the significance of linguistic factors in shaping and perpetuating societal norms and values. This paper provided an in-depth analysis of the subject matter, drawing on how people in the Arab world define, perceive, and refer to “bullying” to illustrate the connection between language and social reality.

Translating the term “bullying” into different languages across different ethnic and cultural groups and countries poses several challenges that must be considered. The primary concern is ensuring the psycholinguistic equivalence of the term “bullying”. It should be noted that certain countries, such as Italy, lack an adequate translation for the English word “bullying”. Additionally, there is no Arabic equivalent for bullying, leading to the ongoing debate about the appropriate term to use and the differences in related concepts [20,33].

Even within the same language, there are variations in the terms used to describe bullying-related behaviour, such as peer harassment or aggression. These differences exist both within and between countries [19]. These can be manifested in Arabic countries, where the same word can mean different things in various countries. Arabic words can also have different meanings depending on the context in which the words are used even within the same country. This is apparent from the results of the current study, where participants from different countries gave different meanings to the bullying scenarios.

Methodological issues also come into play, as research instruments, time frames for survey questions, and the provision of definitions can differ across studies [23,33].

Linguistic issues related to translation and cultural definitions of bullying must also be considered, along with potential measurement invariance related to age and gender differences. Methodological biases, such as translation errors or biases and cultural response style differences, can distort the interpretation of specific items or even the entire instrument [33]. Additionally, it may fail to cover all forms of victimisation, leading to underrepresentation of the construct domain [33].

In addition, caution must be exercised when generalising individual-level constructs to the national or cultural level. The characteristics of bullying or victimisation that apply to individuals within a specific culture or ethnic group may not necessarily represent the entire national or ethnic group. The meaning of bullying or victimisation constructs can vary between individual and cultural levels [37]. It is critical to assess the equivalence of the internal structure in each new ethnic or cultural group where the instrument is applied [19].

Evidence suggests that attitudes toward bullying and violence vary widely across countries. These can be related to cultural differences, linguistic variations [19,20]), or beliefs towards violence, with some supporting physical or corporal punishment for child rearing and some not [38,39].

Cultural variations play a pivotal role in shaping the understanding and perception of bullying within different societies. Consequently, when applying a psychometric tool in a questionnaire-based survey to evaluate bullying in various cultural or ethnic groups, it is imperative to first assess the instrument’s comparability across these groups. This necessity arises due to the cultural specificity of the instrument and the terminology employed to describe bullying.

The significance and features of a particular psychological construct and process can be influenced by cultural systems, leading to differences between different ethnic and national groups [40]. Furthermore, non-methodological factors such as socioeconomic inequality [41] and cultural values (such as individualism-collectivism [42]) can also contribute to country-level variations.

This indicates that cultural values, norms, and beliefs are key factors in shaping diverse perceptions of bullying behaviours. For instance, differences in individualistic and collectivist societies can result in distinct interpretations of the same bullying instrument, leading to varied understandings and perceptions of bullying [42]. In individualistic cultures such as the United States, bullying is often regarded as an individual issue that necessitates intervention on a case-by-case basis. Conversely, collectivist cultures, such as those in many Arab countries, may perceive bullying as a collective problem that requires communal efforts to address.

A cross-cultural study conducted across 75 countries revealed lower overall victimisation rates in individualistic societies, however, a higher prevalence of relational victimisation and a greater ratio of bullies to victims in collectivist societies [42].

The Power Distance Belief framework offers insights into cultural variations in interpreting bullying, and refers to the acceptance of unequal power, status, or wealth distribution within a society or organisation [43]. The theory suggests that societies with high power distance, where there is a significant power difference between individuals and authority figures, may perceive bullying differently compared to societies with low power distance. This can be manifested as a shared value, especially in homogeneous national contexts [43], and as a personal trait at the individual level [44,45] within countries. Core to power distance is the acceptance of inequality, where individuals are more likely to embrace power imbalances [43,46], endorse hierarchical structures, and advocate for subordinates to trust and obey superiors [47]. In the context of bullying, which includes an imbalance of power as one of the definition criteria, individuals with high power distance beliefs accept bullying behaviour as a normal one rather than a negative one but exhibit fewer instances of trust in others. In Arab countries, where power distance tends to be high, bullying may be perceived as more tolerated or normalised within the social hierarchy. As a result, individuals in these societies are less likely to report or intervene in bullying incidents due to perceiving such actions as challenging authority. Accordingly, behaviours considered bullying in some societies may not be classified as such in others, depending on the understanding of the term and its associated connotations.

Additionally, language can significantly affect perceptions of violent acts and bullying behaviours. The choices made in language construction, such as vocabulary selection and discourse patterns, can implicitly endorse or discourage violence. In this study, we found that some terms used may have a positive connotation. Furthermore, linguistic mechanisms, such as metaphorical expressions, can contribute to the normalisation or condemnation of violent behaviour.

In conclusion, cultural differences significantly influence the understanding and perception of bullying. Cultural values and the Power Distance Theory provide theoretical frameworks to explain these differences. By considering these theories, researchers can interpret and analyse the perception of bullying in the Arabic language context and gain a deeper understanding of how cultural factors shape responses and interventions to bullying behaviours.

Research is therefore needed on the relationship between perception, understanding, definition, and actual meaning of the word “bullying” in Arabic-speaking countries. It is also important to investigate these issues among different countries, taking into account the different attitudes, practices, and beliefs. In addition, measurement methods (e.g., questionnaires and interviews) need to be validated more often in Arabic-speaking countries among different ages, genders, and places of residence. In this context, language can shape societal norms, values, and beliefs, constructing a shared reality among individuals within a community. Linguistic frameworks set the boundaries of communication, moulding collective understanding, and influencing the way individuals perceive the world. Thus, exploring language’s impact on the perception and understanding of the term “bullying” is crucial to understanding how societal attitudes towards bullying form and persist.

In relation to gender differences, research has consistently demonstrated that there are indeed gender differences in the conceptualisation of bullying across various cultures and countries [48]. Some societies place significant importance on pride and reputation, particularly for males. This cultural context may lead to different interpretations and manifestations of bullying among males and females. For example, in traditional Arabic societies, males might engage in direct physical aggression as a way to assert dominance and maintain their reputation, while females may resort to indirect methods such as social exclusion or spreading rumours to avoid direct confrontation. In this study, we found that girls were more likely to refer to and identify some of the scenarios as bullying, cyberbullying, gossip, and rumours, while boys defined one of the scenarios as showing strength and control over others. Although these are focus groups with a small number of participants; however, it may indicate how males and females may perceive and define some bullying behaviours differently, and future studies are needed to investigate this further.

A recent study conducted among psychologists in Qatari schools emphasised the importance of psychologists possessing adequate knowledge about bullying and its consequences [49]. The study found that psychologists with higher levels of bullying knowledge and experience demonstrated a better perception of bullying as a problematic behaviour, improved identification of bullying characteristics, greater support for anti-bullying laws, and more bullying guidelines in their workplace. The findings underscore the necessity for practitioner training in addressing bullying in schools. A similar study conducted among psychologists in the UK highlighted the need for further discussion among psychological and legal professionals regarding the definition, characteristics, and impact of bullying and cyberbullying. The study also emphasised the importance of including questions about bullying and cyberbullying in psychological practices, risk assessments, and interventions through education. The study recommended revisions in clinical psychological practices and assessments to reduce bullying prevalence rates, psychological distress, and associated psychopathology that can be comorbid with or emerge as a result of this behaviour [50]. Therefore, based on the current study, a clear definition and perception of bullying will be necessary for the above-mentioned issues.

The study can also inform and be a step-by-step basis for the investigation of bullying definitions and terms in other countries, especially those that do not have a specific word for bullying. It can also inform researchers to investigate the perception and use of the term in various settings, including at home between siblings [3], the workplace, the University etc.

This study has some limitations. First, it includes Arabic-speaking participants from different Arab countries residing in Qatar. These participants’ Arabic use and practises may have been affected by the local dialect of the Gulf country and therefore may not fully represent their original country. Future research is needed to be performed within each Arab country separately. Second, the distribution of gender was not equal in the studies, and therefore there is a need to perform this research amongst ages and genders more comprehensively. Third, the sample size is small and may not be sufficient for understanding between-group differences. Therefore, results need to be treated with caution. However, we emphasise that the results of this study need to be treated qualitatively and give an indication of the need for future studies in the field. Finally, the study did not investigate attitudes and social factors towards the different “bullying” scenarios shown to the participants.

The study has some strengths. This is the first study that investigates bullying terminology in the Arabic language, one of the most spoken languages in the world. It covers various Arabic nationalities as well as different ages and genders. The study used various methodologies to extract the results, including interviews, perceptions, focus groups, and various scenarios to illustrate to the participants the various behaviours that refer to bullying (including various criteria and negative acts) and to elicit valid and reliable responses.

Qualitative as well as quantitative methods are needed more in the Arab world and the Arabic language. This is to establish norms and attitudes, behavioural patterns, gender, age, and cultural differences towards bullying [19]. Further investigations that encompass linguistic analysis of the Arabic language should be conducted to explore the viability of introducing a novel term for bullying within the language. This would serve the purpose of establishing a comprehensive and restructured definition of bullying, aligned with its connotations in English while taking into account cultural values, norms, and beliefs. Such investigations can contribute to the formation of a new term that encompasses the breadth of bullying in the Arabic cultural sphere while simultaneously ensuring linguistic precision and comprehensibility.

## 5. Conclusions

The study highlights the importance of perception, attitudes, and how the word “bullying” is defined in the Arab world. The study is an important step towards defining the word “bullying” in the Arabic language and designing suitable questionnaires and research methods to investigate bullying behaviour in an appropriate way. This is especially important in relation to the positive and/or negative connotations and perceptions that are given to different bullying behaviours.

Previous research conducted in the Arab world has primarily focused on the prevalence of bullying and some associated factors. However, less attention has been given to the specific terminology used to describe bullying in Arabic. It is worth noting that different terms are often used interchangeably, leading to confusion between bullying, violence, aggression, and abuse [33]. This phenomenon has been observed in the present study as well. Using inconsistent terminology can impact the accurate identification of bullying instances, potentially leading to an underestimation of the problem. Consequently, fewer children at risk of experiencing psychological and health issues may be identified. Given the detrimental effects of bullying on both psychological and physical health [3] and academic achievement [4], it is crucial to interpret and perceive this behaviour accurately. This allows for the effective identification and support of children who may be at risk of developing psychological and behavioural problems [50].

Furthermore, it should be noted that the prevalence rates of bullying may vary across different cultures due to the inconsistent use of terminologies and definitions. Therefore, further research is necessary to provide a reliable measurement of the extent of bullying in various cultural contexts.

When determining the precise terminology of bullying, it is essential to consider several factors. Measurement tools should incorporate clear criteria that define bullying, including the repetition of the behaviour, the intention to harm, the imbalance of power, and specific reference points in time. Additionally, they should account for the various types and forms of bullying while considering cultural norms and issues. Previous research has revealed that certain forms of bullying, such as racial and sexual bullying, have been overlooked in some studies in the Arab world due to cultural reasons. Furthermore, the issue of cyberbullying remains understudied despite the widespread use of smartphones and online platforms [33]. Accordingly, there is a need for introducing tailored school-based anti-bullying programmes that are appropriate for specific cultures in order to reduce the negative effects of internet use in general and specifically cyberbullying.

In this context, it is crucial to develop suitable intervention and prevention strategies for addressing bullying. This entails a comprehensive understanding of the term “bullying”, the associated attitudes, its components, perception, and legal ramifications [23,50].

Currently, there is no evidence to indicate that any Arab country has adopted specific anti-bullying policies in schools, except for a few countries with broader behavioural policies [33]. Conversely, in England, it is a legal requirement for all schools to have anti-bullying policies [51]. Introducing a similar legal requirement in Arab countries may incentivise schools to establish their own anti-bullying policies and interventions, ultimately reducing the negative consequences associated with bullying [33,51]. A consistent, unambiguous term for bullying in Arabic can facilitate greater awareness, understanding, and communication regarding the issue among Arabic-speaking populations. This, in turn, can aid in the development of tailored intervention strategies, policies, and educational programmes aimed at preventing and mitigating bullying behaviours. In this study, it is important to note that a subset of the participants consists of migrants and children of migrants. Consequently, interventions aimed at addressing bullying behaviours and enhancing overall well-being must give due consideration to the distinct needs and circumstances of migrants, minorities, and refugees. Inclusiveness and protection of these vulnerable populations should be prioritised to ensure their meaningful engagement in the intervention process [52]. By addressing their unique challenges and experiences, interventions can effectively mitigate their involvement in bullying behaviours and promote their well-being within the broader context of the study.

Overall, addressing the public health implications of bullying in the Arab world requires accurate terminology, training for practitioners, including teachers and psychologists, parental involvement, and the implementation of effective policies and interventions in schools.

## Figures and Tables

**Table 1 ijerph-21-00364-t001:** Demographic data for Study 1 (*N*: 36).

	Gender		Age	Total
Nationality	Male (%)	Female (%)	Means (SD)	*n* (%)
Egyptian	33.3%	66.7%	32.57 (14.60)	9 (27.0%)
Qatari	25.0%	75.0%	18.86 (6.64)	12 (32.4%)
Syrian	0.0%	100.0%	22.67(5.03)	4 (10.8%)
Other Arab nationalities *	45.5%	54.5%	30.80 (8.06)	11 (29.8%)

* Other Arab nationalities represented countries that were found in no more than 2 participants in the total sample: Palestine: 2; Tunisia: 1; Jordan: 2; Somalia: 1; Iran: 1; Bahrain: 1; Yemen: 2; Iraq: 1.

**Table 2 ijerph-21-00364-t002:** Seven-item word list describing the characteristics of bullying presented to participants.

Chosen Term		
Arabic Term	Transliteration	Translation
تسلط	Tassalot’	Dominance
تعدي	Ta’adi	Trespass
مضايقة	Modayaqah	Harassment
أذى	Az’a	Harm
بلطجة	Baltag’ah	Gangsterism
تنمر	Tanammor	Bullying (Resembling the tiger)
مشاكسة	Moshakasah	Teasing

**Table 3 ijerph-21-00364-t003:** Demographic data for Study 2 (*N*: 117).

	Type of School		Age	Total
Nationality	International *n* (%)	Independent *n* (%)	Means (SD)	*n* (%)
Egyptian	13 (72.2)	5 (27.8)	14.4 (1.46)	18 (15.4)
Qatari	15 (56.8)	28 (43.2)	14.6 (1.32)	43 (36.7)
Palestinian	3 (27.3)	8 (72.7)	13.5 (0.89)	11 (9.4)
Syrian	0 (0.0)	5 (100)	14.8 (0.45)	5 (4.3)
Other Nationalities	30 (75)	10 (25) *	14.7 (1.28)	40 (34.2)

* Other Arab nationalities: Jordan: 3; Saudi Arabia: 3; Sudan: 2; Somalia: 1; Lebanon: 1.

**Table 4 ijerph-21-00364-t004:** The terms that participants used to describe the behaviour being depicted across the three scenarios.

Scenarios *								
Scenario 1			Scenario 2			Scenario 3		
Arabic Term Used	English Translation	*n* (%)	Arabic Term Used	English Translation	*n* (%)	Arabic Term Used	English Translation	*n* (%)
سرقة	Theft	6 (16.6)	غيرة	Jealousy	8 (19.4)	عدائي	Aggression	6 (16.6)
غير طبيعي	Abnormal	2 (5.6)	نشر إشاعات	Spreading Rumours	4 (11.1)	عنف	Violence	4 (11.1)
غيرة	Jealousy	6 (16.6)	تنمر إلكتروني	Cyber-bullying	3 (5.5)	زعيم البلطجة	Leader of a Gangster	1 (2.8)
لفت الإنتباه	Attention seeking	4 (11.1)	بلطجة	Gangster	1 (2.8)	لعوزة **	Reprimand	2 (5.6)
بلطجة	Gangster	2 (5.6)	أذى	Harming	3 (13.8)	غلاسة	Silliness	1 (2.8)
كراهية	Hatred	1 (2.8)	كراهية	Hatred	1 (2.8)	أذى	Harming	1 (2.8)
غَلاَسَة	Silliness	1 (2.8)	موقف مخيف	Scary situation	1 (2.8)	يدافع عن نفسه	Self-defence	1 (2.8)
تنمر	Bullying (Resembling the tiger)	1 (2.8)	تقليد لفعل سيء رآه في البيت	Imitating bad behaviour observed at home	1 (2.8)	تنمر جسدي	Physical Bullying	2 (5.6)
أفعال صبيانية	Childish acts	1 (2.8)	إستظراف	Trying to be funny/joke	1 (2.8)	تصرف طبيعي	Normal Behaviour	2 (5.6)
مضايقة	Annoying	1 (2.8)	تسلط	Forcing Control (Dominance)	1 (2.8)	سيطرة	Controlling	2 (5.6)
تهجم	Attacking	1 (2.8)	للشهرة	To be Famous	1 (2.8)	غضب	Anger	2 (5.6)
ثقة زائدة	Over Trust	1 (2.8)	شَر	Evil	1 (2.8)	مزاح	Joke	1 (2.8)
إنتقام	Revenge	1 (2.8)	جريمة إجتماعية	Social Crime/bully	2 (5.6)	رد فعل	Reaction	2 (5.6)
حالة سيئة	Bad Situation	1 (2.8)	سرقة	Theft	1 (2.8)	غَلاَسَة	Silliness	1 (2.8)
تقييم خاطىء	Lack of Judgement	1 (2.8)	-	No Answer	5 (13.8)	فَتْوَنَة ***	Gangsterism	2 (5.6)
رد فعل	Reaction	1 (2.8)	-	-	-	تهجم	Attacking	1 (2.8)
-	No Answer	5 (13.9)	-	-	-	-	No Answer	6 (16.6)

* Scenarios: (1) Every day at school, Reem steals Heba’s chocolate bar taken off her from her lunchbox. She also sometimes takes her money; (2) He doesn’t take things from any other people in the class; and Mohamed and Haitham regularly and anonymously spread false stories about one of their colleagues in class through Facebook. They do this with the intention of harming him; (3) Mohammed punched Faisal in the face during the lunch break at school, leaving him with a bleeding mouth. ** A word usually used in the Gulf countries. *** Originally, the word “Fatwana” had a positive connotation of being valiant and helping the weak. Gradually, it started to have a negative meaning, referring to being a gang member and “charging” people for their “protection” services. This is similar to the word بلطجي “balTag’i”, someone using his physical power to gain money illegally.

**Table 5 ijerph-21-00364-t005:** Frequency of terms chosen from a 7-item word list used to describe each of the three scenarios at the end of the focus group.

Chosen Term			Scenarios (*n*; %)			
Arabic Term	Transliteration	Translation	1	2	3	Total (*n*; %)
تسلط	Tassalot’	Dominance	12 (31.6)	2 (5.0)	1 (2.6)	15 (39.2)
تعدي	Ta’adi	Trespass	10 (26.2)	11 (27.5)	7 (18.0)	28 (71.7)
مضايقة	Modayaqah	Harassment	6 (15.8)	4 (10.0)	5 (12.8)	15 (38.6)
أذى	Az’a	Harm	3 (7.9)	16 (40.0)	8 (20.5)	27 (68.4)
بلطجة	Baltag’ah	Gangsterism	3 (7.9)	2 (5.0)	8 (20.5)	13 (33.4)
تنمر	Tanammor	Bullying (Resembling the tiger)	2 (5.3)	2 (5.0)	5 (12.8)	9 (23.1)
مشاكسة	Moshakasah	Teasing	2 (5.3)	3 (7.5)	5 (12.8)	10 (22.6)
** Overall (*N*) **			38 (100)	40 (100)	39 (100)	117

**Table 6 ijerph-21-00364-t006:** The frequency of the terms chosen from the seven-item word list for each of the three scenarios across the four nationality groups in Study 1.

Type of Scenario	Arabic Chosen Term and Transliteration	Translation to English	Arabic Nationality (*n*; %)			
			Egyptian	Qatari	Syrian	Other
** Scenario 1 **	تسلط—Tassalot’	Dominance	3 (33.3)	3 (25.0)	2 (50.0)	3 (27.3)
	تعدي—Ta’adi	Trespass	-	4 (33.3)	1 (25.0)	4 (36.4)
	مضايقة—Modayaqah	Harassment	3 (33.3)	3 (25.0)	-	1 (9.1)
	أذى—Az’a	Harm	1 (11.1)	1 (8.3)	-	-
	بلطجة—Baltag’ah	Gangsterism	1 (11.1)	-	1 (25.0)	1 (9.1)
	No Answer		1 (11.1)	-	-	-
** Scenario 2 **	تسلط—Tassalot’	Dominance	1 (11.1)	-	1 (25.0)	1 (9.1)
	تعدي—Ta’adi	Trespass	1 (11.1)	-	-	1 (9.1)
	مضايقة—Modayaqah	Harassment	1 (11.1)	-	-	-
	أذى—Az’a	Harm	2 (22.2)	5 (41.7)	3 (75.0)	6 (54.5)
	تنمر—Tanammor	Bullying (Resembling the tiger)	1 (11.1)	1 (8.3)	-	-
	مشاكسة—Moshakasah	Teasing	2 (22.2)	3 (25.0)	-	-
	بلطجة—Baltag’ah	Gangsterism	-	1 (8.3)	-	1 (9.1)
	No Answer		1 (11.1)	2 (16.7)	1 (25.0)	2 (18.2)
** Scenario 3 **	تسلط—Tassalot’	Dominance	-	-	-	3 (27.3)
	تعدي—Ta’adi	Trespass	1 (11.1)	3 (25.0)	1 (25.0)	4 (36.4)
	مضايقة—Modayaqah	Harassment	-	-	2 (50.0)	1 (9.1)
	أذى—Az’a	Harm	-	4 (33.3)	-	-
	تنمر—Tanammor	Bullying (Resembling the tiger)	-	1 (8.3)	-	-
	مشاكسة—Moshakasah	Teasing	1 (11.1)	3 (25.0)	-	-
	بلطجة—Baltag’ah	Gangsterism	5 (55.6)	1 (8.3)	1 (25.0)	1 (9.1)

**Table 7 ijerph-21-00364-t007:** Frequency of the terms used to describe the behaviours in the 7 scenarios from participants in international schools in Study 2 (*N* = 71).

Terms Used	Scenarios * (*n*; %)						
	1	2	3	4	5	6	7
Bullying	29 (47.5)	12 (19.7)	4 (6.6)	17 (27.9)	-	3 (4.9)	4 (6.6)
Mockery	1 (1.6)	-	-	-	-	-	-
Gossip	5 (8.2)	-	-	-	-	23 (37.7)	-
Jealousy	3 (4.9)	3 (4.9)	4 (6.6)	-	-	-	5 (8.2)
Making fun	4 (6.6)	-	-	-	-	-	-
Name Calling	2 (3.3)	-	-	-	-	-	-
Bad Attitude	2 (3.3)	-	-	-	-	-	-
Cyberbullying	-	18 (29.5)	17 (27.9)	-	-	-	-
Hurting	1 (1.6)	-	-	1 (1.6)	-	1 (1.6)	-
Insult	1 (1.6)	1 (1.6)	-	-	-	-	-
Hatred	-	-	6 (9.8)	-	-	-	-
Physical Bullying	-	-	-	14 (23.0)	-	-	-
Violence	-	-	-	1 (1.6)	-	-	-
Threat	-	-	-	2 (3.3)	-	-	-
Assault	-	-	-	-	-	-	-
Joke	-	-	-	-	16 (26.2)	-	-
Reaction	-	-	-	-	8 (13.1)	-	-
Misunderstanding	-	-	-	-	4 (6.6)	-	-
Anger	-	-	-	-	3 (4.9)	-	-
Immature	-	-	-	-	-	2 (3.3)	-
Exclusion	-	-	-	-	-	-	3 (4.9)
Left out	-	-	-	-	-	-	6 (9.8)
Childish	-	-	-	-	-	-	2 (2.8)

* Scenarios: (1) A group of girls are making fun of one of the girls at school; they often call her bad names; (2) Three friends in the same class repeatedly send Faisel text messages calling him names (impolite). Faisal is upset by this but does not say anything to anyone; (3) A boy/girl keeps receiving a message on his/her computer from an unknown person. The messages are often strange and say things such as ‘I hate you’; (4) Over the last three weeks, a student has been repeatedly hit by other students; (5) Hasan once joked with Ahmad, but Ahmad got angry and punched him in the face; (6) A group of friends are telling untrue stories about one of their colleagues; they have done that many times; (7) Mohamed and his friends never let Naif play with them.

**Table 8 ijerph-21-00364-t008:** The frequency of the terms used to describe the behaviours in the 7 scenarios from participants in independent schools in Study 2 (*N* = 47).

Arabic Term and Transliteration	English Translation	Scenarios * (*n*; %)						
		1	2	3	4	5	6	7
إستهزاء—Istihzaa	Making Fun	29 (51.8)	3 (5.4)	3 (5.4)	-	-	3 (6.3)	-
سخرية—Sokhriyah	Mocking	16 (28.6)	-	-	-	-	-	-
إهتمام عدم—A’adam Ihtimam	Carelessness	6 (10.7)	5 (8.9)	-	-	-	1 (1.7)	-
سب—Sab	Swearing	4 (7.1)	7 (12.5)	-	-	-	-	-
تنمر—Tanammor	Bullying (Resembling the tiger)	3 (5.4)	1 (1.7)	-	5 (8.9)	-	-	1 (1.7)
مضايقة—Modayaqah	Harassment	1 (1.7)	4 (7.1)	2 (3.6)	-	-	-	-
سيء سلوك—Solok sayea’	Bad behaviour	4 (7.1)	9 (16.1)	9 (16.1)	6 (10.7)	-	-	-
إعتداء لفظي—Ea’teda lafd’ee	Verbal assault	-	1 (1.7)	-	-	-	-	-
للمشاعر جرح—Jarih lil-mashaeir	Hurt the feelings	-	3 (5.4)	4 (7.1)	-	-	-	-
كراهية—Karahiyah	Hatred	-	-	6 (10.7)	-	-	4 (7.1)	18 (32.1)
كذب—Kaz’ib	Lie	-	-	3 (5.4)	-	-	8 (14.3)	
إستقواء—Istiqwaa’	Perception of strength and control over others	-	-	-	7 (12.5)		-	-
إعتداء—Ia’tidaa	Assault	-	1 (1.7)		6 (10.7)	6 (10.7)	-	-
تكبر—Takabor	Arrogance	-	-	-	3 (5.4)		-	4 (7.1)
عنف—A’onf	Violence	-	-	-	2 (3.6)	3 (5.4)	-	-
ثقيل مزح—Mazih Thaqeel	Heavy joking	-	-	-	-	19 (33.9)	-	-
غاضب—Ghad’eb	Angry	-	-	-	-	12 (21.4)	-	-
ضرب—D’arb	Hitting	-	1 (1.7)	-	-	5 (8.9)	-	-
غيبة—Ghiba	Gossip	-	-	-	-	-	7 (12.5)	-
غيرة—Gherah	Jealousy	-	-	1 (1.7)	-	-	5 (8.9)	-
تشويه—Tashweeh	Reputation distortion	-	-	-	-	-	4 (7.1)	-
أنانية—Ananeyah	Selfishness	-	-	-	-	-	-	8 (14.3)
حقد—H’ik’d	Grudge	-	-	-	-	-	-	2 (3.6)
ظالم—D’alem	Oppressor (unjust)	-	-	1 (1.7)	-	-	1 (1.7)	6 (10.7)

* Scenarios: (1) A group of girls are making fun of one of the girls at school; they often call her bad names; (2) Three friends in the same class repeatedly send Faisel text messages calling him names (impolite). Faisal is upset by this but does not say anything to anyone; (3) A boy/girl keeps receiving a message on his/her computer from an unknown person. The messages are often strange and say things such as ‘I hate you’; (4) Over the last three weeks, a student has been repeatedly hit by other students; (5) Hasan once joked with Ahmad but Ahmad got angry and punched him in the face; (6) A group of friends are telling untrue stories about one of their colleagues; they have done that many times; (7) Mohamed and his friends never let Naif play with them.

## Data Availability

Data is available upon request from the corresponding author.

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
