# Peer review of "Bullying in the Arab World: Definition, Perception, and Implications for Public Health and Interventions"

_ijerph, 2024, doi:10.3390/ijerph21030364_

Round 1
Reviewer 1 Report
Comments and Suggestions for Authors
The advantage of this paper is the choice of the research subject. Perception of bullying in the Arabic Culture is a very poorly researched issue so far. From this point of view, this project is innovative in the sphere of applied methodological solutions. The practical dimension of such research findings is also worth emphasizing. It is difficult to design preventive actions without knowledge of how bullying is understood in culture. It is the factor which determines the processes of socialization of young generations. Therefore, research projects carried out in this way have great educational potential.
The advantages of this research are primarily:
1. Good embedding of the project in the research conducted so far.
2. The importance attached to the precision of the research project - in particular, the use of focus groups as a method of gathering empirical data
However, I think that for the sake of the academic soundness of the findings, it is worth rethinking a few detailed issues.
1. Considering the possibility of including references to theories explaining the sources of cultural differences in the understanding of bullying. It is not enough, especially in qualitative research, to refer to the results of research conducted so far. It is necessary to create a theoretical framework for the interpretation of the obtained results
2. Applied methods of empirical data analysis favors quantitative (statistical) methods. The method of using content analysis is insufficient. For the sake of the quality of the research results, it is worth rethinking the issues of balance between quantitative and qualitative analysis. The quantitative analysis does not bring much here - the reason is the lack of representativeness of research groups. The results of qualitative research (e.g. by referring to discourse analysis) may lead to an answer to the question of to what extent language (which is, after all, a basic component of culture) creates social reality – in particular tolerance to acts of violence.
Despite the above-mentioned doubts, I state that the research presented in the article meets the criteria of reliability and validity - hence I recommend its publication
Author Response
Dear Reviewer,
We would like to thank the reviewer for their valuable comments and insights. The comments have improved the manuscript. We outline below the reply for each raised point with out reply:
Comments:
The advantage of this paper is the choice of the research subject. Perception of bullying in the Arabic Culture is a very poorly researched issue so far. From this point of view, this project is innovative in the sphere of applied methodological solutions. The practical dimension of such research findings is also worth emphasizing. It is difficult to design preventive actions without knowledge of how bullying is understood in culture. It is the factor which determines the processes of socialization of young generations. Therefore, research projects carried out in this way have great educational potential.
The advantages of this research are primarily:
- Good embedding of the project in the research conducted so far.
- The importance attached to the precision of the research project - in particular, the use of focus groups as a method of gathering empirical data
However, I think that for the sake of the academic soundness of the findings, it is worth rethinking a few detailed issues.
- Considering the possibility of including references to theories explaining the sources of cultural differences in the understanding of bullying. It is not enough, especially in qualitative research, to refer to the results of research conducted so far. It is necessary to create a theoretical framework for the interpretation of the obtained results
Our reply: We have added more details about the theoretical framework in the discussion section. Please see page 21.
Comment:
- Applied methods of empirical data analysis favors quantitative (statistical) methods. The method of using content analysis is insufficient. For the sake of the quality of the research results, it is worth rethinking the issues of balance between quantitative and qualitative analysis. The quantitative analysis does not bring much here - the reason is the lack of representativeness of research groups. The results of qualitative research (e.g. by referring to discourse analysis) may lead to an answer to the question of to what extent language (which is, after all, a basic component of culture) creates social reality – in particular tolerance to acts of violence.
Our reply: We have added more details about this. Please see the discussion, pages 19-23.
Comment: Despite the above-mentioned doubts, I state that the research presented in the article meets the criteria of reliability and validity - hence I recommend its publication.
Thank you and best wishes.
Reviewer 2 Report
Comments and Suggestions for Authors
According to the authors of this very interesting paper, their study is the first to attempt to investigate the understanding of a variety of bullying-related terms in Arabic-speaking children and adults. In addition, it is also the first study to check for differences in the terms used in the Arabic language.
To achieve their objectives, two studies were conducted to examine (1) How Middle Eastern Arabic-speaking participants describe and perceive bullying behavior and (2) How Middle Eastern participants describe bullying behavior using Arabic and English terms.
In the introductory section they provide important information about bullying among school children in different countries, as well as the different terms used to define bullying in different languages.
For the needs of the first study, 36 people were recruited, from different Arab nationalities, all residents of Qatar. Qualitative data collection was used to understand the terms the participant uses to describe behaviors that represent the characteristics of bullying.
Regarding the second study, 117 participants were recruited from various schools representing different Arab nationalities. This study consisted of 7 scenarios, 6 of which represented the characteristics of bullying behavior, and participants were asked to verbalize the term describing the behavior presented in each scenario.
Content analysis, qualitative analysis and descriptive statistics have been used for the needs of the two studies.
Both the results and the discussion section are developed to a satisfactory level.
I believe the paper could be published as is, although I have 2 minor comments: a) tables 6 & 7 have many cells with 0 answers and I wonder if there is another way to present it so as not to disturb the readers, b) I was under the impression that the United States is not part of the "Pacific Rim countries".
Author Response
Dear Reviewer,
We would like to thank the reviewer for their valuable comments and insights. The comments have improved the manuscript. We outline below the reply for each raised point with out reply:
Comments:
Reviewer 2
According to the authors of this very interesting paper, their study is the first to attempt to investigate the understanding of a variety of bullying-related terms in Arabic-speaking children and adults. In addition, it is also the first study to check for differences in the terms used in the Arabic language.
To achieve their objectives, two studies were conducted to examine (1) How Middle Eastern Arabic-speaking participants describe and perceive bullying behavior and (2) How Middle Eastern participants describe bullying behavior using Arabic and English terms.
In the introductory section they provide important information about bullying among school children in different countries, as well as the different terms used to define bullying in different languages.
For the needs of the first study, 36 people were recruited, from different Arab nationalities, all residents of Qatar. Qualitative data collection was used to understand the terms the participant uses to describe behaviors that represent the characteristics of bullying.
Regarding the second study, 117 participants were recruited from various schools representing different Arab nationalities. This study consisted of 7 scenarios, 6 of which represented the characteristics of bullying behavior, and participants were asked to verbalize the term describing the behavior presented in each scenario.
Content analysis, qualitative analysis and descriptive statistics have been used for the needs of the two studies.
Both the results and the discussion section are developed to a satisfactory level.
I believe the paper could be published as is, although I have 2 minor comments: a) tables 6 & 7 have many cells with 0 answers and I wonder if there is another way to present it so as not to disturb the readers, b) I was under the impression that the United States is not part of the "Pacific Rim countries".
Our reply: Thank you for the positive comments, that is very helpful. We have changed tables 6 and 7 to make them more presentable. We did the same with the rest of the tables as well. We have also changed the wording of the sentence about the Pacific Rim countries.
Thank you and best wishes.
Reviewer 3 Report
Comments and Suggestions for Authors
The manuscript entitled “Definition and Perception of bullying in the Arabic Culture: Understating the terminology and perception used to describe bullying acts in the Arab world” is an interesting and valuable addition to the literature on cross-cultural differences in bullying. In two separate studies, the authors’ research offers a detailed conceptualization of how views of bullying may overlap or differ among diverse Arab nationalities. The manuscript deserves publication after a few concerns are addressed.
In the introduction, the authors focus on the different words that in the Arabic language are often used to refer to bullying. Are there similar or dissimilar linguistic distinctions in other languages? Most importantly, words do not appear in isolation. Is there any evidence that the different words used in the Arabic language to refer to bullying are inserted into different contextual frames and thus serve different pragmatic purposes?
The authors need to present their hypotheses before the method section. Each hypothesis is to have a rationale. For instance, do the authors expect differences? If so, what are the predictions based on the extant literature?
In the method section of Study 1, the age range of the participants is unusually broad and gender is distributed unevenly across nationalities. Please explain the inclusion and exclusion criteria. Is there a reason to assume that the age of the participants does not matter in the conceptualization of bullying? The sample size is also exceptionally small. Is this sample size sufficient for understanding between-group differences? The authors’ reliance on convenience sampling and a small sample of participants may question the quality of the data collected. Namely, is there room for biases in the results of the study that can be attributed to methodological issues?
The category “other nationalities” needs to be clarified with detailed information about the inclusion of participants.
How were the scenarios constructed and validated? Please describe the pilot study that preceded the current research. The goal here is to ensure that scholars can replicate the study.
In the organization of the data into separate semantic categories, is there a way to compute an inter-rater reliability score?
The discussion section is unsatisfactory for several reasons. First and foremost, the authors may more comprehensively examine not only the existence of different terms for bullying but also their likely use in different pragmatic contexts. Second, the small and heterogeneous participant group of Study 1 may have offered a skewed window into the conceptualization of bullying. Potential differences involving age groups may need to be considered seriously. Third, how do the data of the current study compare and contrast with the data of other studies examining different languages? Forth, one of the limitations of the study is the unbalanced ratio of male and female participants. Are there gender differences in the conceptualization of bullying in other languages? Are there reasons to expect gender differences in the Arabic language too? Fifth, the authors need to provide a comprehensive overview of the public health implications of bullying in the Arab world and across the globe.
Author Response
Dear Reviewer,
We would like to thank the reviewer for their valuable comments and insights. The comments have improved the manuscript. We outline below the reply for each raised point with out reply:
Comments:
The manuscript entitled “Definition and Perception of bullying in the Arabic Culture: Understating the terminology and perception used to describe bullying acts in the Arab world” is an interesting and valuable addition to the literature on cross-cultural differences in bullying. In two separate studies, the authors’ research offers a detailed conceptualization of how views of bullying may overlap or differ among diverse Arab nationalities. The manuscript deserves publication after a few concerns are addressed.
In the introduction, the authors focus on the different words that in the Arabic language are often used to refer to bullying. Are there similar or dissimilar linguistic distinctions in other languages?
Our reply: We have elaborated more about the effect of language in the discussion section Please see pages 19-23.
Most importantly, words do not appear in isolation. Is there any evidence that the different words used in the Arabic language to refer to bullying are inserted into different contextual frames and thus serve different pragmatic purposes?
Our reply: We have added some examples beneath table 3 and more elaboration about this in the discussion section, please see pages 19-23.
The authors need to present their hypotheses before the method section. Each hypothesis is to have a rationale. For instance, do the authors expect differences? If so, what are the predictions based on the extant literature?
Our reply: We have added the hypothesis. Please see page 3.
In the method section of Study 1, the age range of the participants is unusually broad and gender is distributed unevenly across nationalities. Please explain the inclusion and exclusion criteria. Is there a reason to assume that the age of the participants does not matter in the conceptualization of bullying?
Our reply: We have mentioned this as a limitation and explained in the methodology. Please see page 4. Follow up studies could potentially explore differences in relation to age based on the tools of the current study. This is now mentioned in the discussion section.
The sample size is also exceptionally small. Is this sample size sufficient for understanding between-group differences? The authors’ reliance on convenience sampling and a small sample of participants may question the quality of the data collected. Namely, is there room for biases in the results of the study that can be attributed to methodological issues?
Our reply: This was added as a limitation of the study in the discussion. Please see pages 21-22
The category “other nationalities” needs to be clarified with detailed information about the inclusion of participants.
Our reply: The details has been added to tables 1 and 2.
How were the scenarios constructed and validated? Please describe the pilot study that preceded the current research. The goal here is to ensure that scholars can replicate the study.
Our reply: This has been explained in more details in the methodology section page 5.
In the organization of the data into separate semantic categories, is there a way to compute an inter-rater reliability score?
Our reply: This was commented on page 8.
The discussion section is unsatisfactory for several reasons. First and foremost, the authors may more comprehensively examine not only the existence of different terms for bullying but also their likely use in different pragmatic contexts.
Our reply: This has been now discussed and more discussion points has been elaborated on including language, cultural issues, theoretical framework etc. Please see pages 19-23.
Second, the small and heterogeneous participant group of Study 1 may have offered a skewed window into the conceptualization of bullying. Potential differences involving age groups may need to be considered seriously.
Our reply: Due to the small number of each age group and the number of questions and answers we couldn’t perform age differences analysis. We have put this as a limitation in the discussion and recommended that future research can replicate our study to investigate age and gender differences in more depth.
Third, how do the data of the current study compare and contrast with the data of other studies examining different languages?
Our reply: We gave some examples in the discussion section. Please see pages 19-23.
Forth, one of the limitations of the study is the unbalanced ratio of male and female participants. Are there gender differences in the conceptualization of bullying in other languages? Are there reasons to expect gender differences in the Arabic language too?
Our reply: We have added this to the discussion section. Please see page 21.
Fifth, the authors need to provide a comprehensive overview of the public health implications of bullying in the Arab world and across the globe.
Our reply: We have elaborated on this at the conclusion section. Please see pages 22-23.
Thank you and best wishes.
Round 2
Reviewer 3 Report
Comments and Suggestions for Authors
The manuscript entitled “Bullying in the Arab World: Definition, Perception and Implications for Public Health and Interventions” has been revised by the authors. In my modest opinion, the authors have adequately addressed the issues mentioned in the earlier review of the manuscript. My only remaining concern pertains to the area of hypothesis testing. I would encourage the authors to derive the hypotheses from each of the research questions and state them unambiguously. However, if they prefer to abstain from making predictions, the current state of the extant literature may offer a reasonable justification.
Author Response
Dear Reviewer,
Thank you for your valuable comments. We have addressed the comment about the hypotheses and we have also improved the section at the end of the introduction. We have added further information on pages 3-4, lines 130-162.
Best wishes,
Corresponding author,
Muthanna Samara